# The Effects of Eplerenone on the Circadian Blood Pressure Pattern and Left Ventricular Hypertrophy in Patients with Obstructive Sleep Apnea and Resistant Hypertension—A Randomized, Controlled Trial

**DOI:** 10.3390/jcm8101671

**Published:** 2019-10-13

**Authors:** Beata Krasińska, Szczepan Cofta, Ludwina Szczepaniak-Chicheł, Piotr Rzymski, Tomasz Trafas, Lech Paluszkiewicz, Andrzej Tykarski, Zbigniew Krasiński

**Affiliations:** 1Department of Hypertension, Angiology and Internal Disease, Poznan University of Medical Sciences, 61-848 Poznan, Poland; l.vinnie@wp.pl (L.S.-C.); tykarski@o2.pl (A.T.); 2Department of Pulmonology, Allergology and Lung Oncology, Poznan University of Medical Sciences, 61-001 Poznan, Poland; s.cofta@gmail.com (S.C.); tomasztrafas@wp.pl (T.T.); 3Department of Environmental Medicine, Poznan University of Medical Sciences, 60-806 Poznan, Poland; rzymskipiotr@ump.edu.pl; 4Bad Oeynhausen, Heart and Diabetes Center NRW, Ruhr-University of Bochum, 32545 Bad Oeynhausen, Germany; lpalusz@poczta.onet.pl; 5Department of Vascular, Endovascular Surgery, Angiology and Phlebology, Poznan University of Medical Sciences, 61-848 Poznan, Poland; zbigniew.krasinski@gmail.com

**Keywords:** obstructive sleep apnea, resistant hypertension, non-dippers, mineralocorticoid receptor antagonists, eplerenone, left ventricular hypertrophy

## Abstract

The obstructive sleep apnea (OSA) is highly associated with various significant cardiovascular outcomes such as resistant hypertension (RAH). Despite this, as of now the relationship between high night-time blood pressure (BP) and left ventricular hypertrophy (LVH) in patients with OSA and RAH is not fully understood. The aim of this study was to assess the influence of the addition of eplerenone to a standard antihypertensive therapy on parameters of 24-h ambulatory blood pressure measurement (ABPM) as well as on the results of echocardiography and polysomnography in patients with OSA and RAH. The patients were randomly assigned to one of the two study groups: the treatment group, receiving 50 mg/d eplerenone orally for 6 months (*n* = 51) and the control group, remaining on their standard antihypertensive therapy (*n* = 51). After that period, a significant reduction in the night-time BP parameters in the treatment group including an increased night blood pressure fall from 4.6 to 8.9% was noted. Additionally, the number of non-dipper patients was reduced by 45.1%. The treatment group also revealed a decrease in left ventricular hypertrophy and in the apnea–hypopnea index (AHI) with a positive correlation being observed between these two parameters. This study is the first to report the improvement of the circadian BP profile and the improvement of the left ventricle geometry in patients with OSA and RAH following the addition of selective mineralocorticoid receptor antagonists to antihypertensive therapy.

## 1. Introduction

The prevalence of obstructive sleep apnea (OSA) appears to increase, and currently affects over 20% of men and over 15% of women [1,2]. It is known that OSA is highly associated with various significant cardiovascular outcomes such as resistant hypertension (RAH) [3,4,5]. This specific group of patients is burdened with a very high risk of cardiovascular complications and reveals an extensive organ damage, primarily in the form of left ventricular hypertrophy (LVH) [6]. In patients with OSA, frequent episodes of hypoxemia, hypercapnia, and arousals during sleep result in the repeated activation of the sympathetic nervous system with increased secretion of catecholamines. Moreover, the episodes of respiratory distress increase aldosterone serum concentration, resulting in sodium and water retention, leading to elevated blood pressure (BP). An increased aldosterone level also stimulates synthesis of collagen, promotes stiffening of the arterial wall, myocardial fibrosis with heart muscle remodeling, and contributes to the development of LVH.

Several studies, including the Sleep Heart Health Study, confirmed that severe OSA is associated with high prevalence of concentric hypertrophy through sympathetic activation and vasoconstriction [7,8,9]. Other studies evidenced that patients with RAH, with concomitant OSA and LVH, are at high risk of cardiovascular complications [10,11]. As demonstrated, an increase in aldosterone concentration causes the development of resistant hypertension, deterioration of the circadian BP profile, and LVH. Therefore, it seems that the most promising class of antihypertensive drugs would be represented by diuretics—in particular, mineralocorticoid receptor antagonists (MRAs), based on their influence on the severity of OSA and BP, as presented previously in literature [10,12]. Furthermore, the studies using 24-h automated blood pressure monitoring (ABPM) showed that patients with OSA had higher prevalence of an unfavorable circadian BP pattern compared with subjects not suffering from this disorder. In this group of patients, the non-dippers’ and risers’ profile was more common [13,14]. However, the correlation between the disturbances in the circadian BP profile and the incidence of organ damage is still being debated. Cuspidi et al. analyzed 26 studies published between 2000 and 2009 with a total of 3877 patients under observation. A positive correlation between non-dipping status and LVH was shown in 17 of them, whereas in the remaining 9 studies no such association was proven [15,16].

The present randomized, controlled clinical trial aimed to investigate whether addition of eplerenone to a standard antihypertensive treatment could result in changes in ABPM as well as polysomnographic and echocardiographic parameters in patients with OSA and RAH.

## 2. Experimental Section

### 2.1. Study Design

Patients enrolled in the study underwent two visits in accordance with the prescribed treatment schedule (Figure 1).

On the first—qualifying visit—the patients had been admitted to the ward where the laboratory tests, weight, body mass index (BMI) and neck circumference measurement, abdominal ultrasound examination, abdominal CT scan, Doppler ultrasound of renal arteries, office BP (3×/24 h), ABPM, echocardiography, and polysomnography were conducted. The study was conducted in the open-label fashion. The patients were randomly assigned, in a blinded fashion (a sealed envelope principle) to one of the two study groups receiving 50 mg of eplerenone per day in a single oral morning dose (*n* = 51) or remaining on their standard antihypertensive therapy with no placebo (*n* = 51). None of the control subjects received either eplerenone or spironolactone before or during the study. All patients in both groups were advised to follow rules of nonpharmacological treatment of AH (decrease in the amount of sodium in diet, restrictions in the amount of alcohol, Mediterranean diet, and increase in physical activity were suggested) in accordance with the ESH/ESC guidelines for management of arterial hypertension. After 6 months, Visit 2 was held, during which the examinations from the initial visit were repeated. The patients were also instructed to bring the boxes with pharmaceuticals to Visit 2, and based on the number of pills taken during the time of treatment, the patient’s compliance was checked. The patients followed the recommendations and at least 80% of the recommended doses were taken.

### 2.2. Eligibility Criteria for Inclusion, Exclusion Criteria, and Recruitment

A total of 102 patients (59 men and 43 women) with diagnosed RAH and moderate or severe OSA were enrolled in the study, which was conducted in years 2014–2018 at the University of Medical Sciences in Poznan, Poland. The patients with previously diagnosed RAH and with suspected OSA (medical history, the Epworth Sleepiness Scale) were referred from an outpatient clinic to a hospital ward (Department of Hypertension, Angiology and Internal Medicine). After laboratory and imaging tests, as well as after confirming RAH based on ABPM, 147 patients were referred to the Department of Pulmonology, Allergology and Lung Oncology to undergo polysomnography. Patients in whom moderate (apnea–hypopnea index (AHI) 16–30/h) or severe OSA (AHI > 30/h) had been confirmed (102 patients) were passed on again to the Department of Hypertension, where randomization procedures were carried out. RAH was diagnosed when it was not possible to achieve the target values of BP (<140/90 mmHg) in spite of using at least three antihypertensive agents (including a diuretic) at maximum doses [17]. The enrolled patients were taking on average 4 antihypertensive medications, including diuretics (100% of patients), angiotensin-converting enzyme inhibitors (54% of patients), angiotensin II receptor antagonists (45.2%), calcium antagonists (83.9%), β-blockers (77.4%), and α-blockers (22.6%). The exclusion criteria were as follows: secondary hypertension (other than primary hyperaldosteronism), myocardial infarction, stroke within 6 months before the study, congestive heart failure with NYHA grade III–IV, chronic kidney disease (glomerular filtration rate (GFR) < 60 mL/min), active addiction to alcohol or psychoactive substances, and active cancer disease.

### 2.3. Ethical Aspects and Informed Consent

The study was approved by the Local Bioethical Committee at Poznan University of Medical Sciences (Approval No. 565/14), and registered at ClinicalTrials.gov (ID: NCT03206944). All patients gave an informed and written consent to participate in the study.

### 2.4. Measurements

#### 2.4.1. Blood Pressure

During each visit, all patients underwent BP measurements performed three times at rest in supine position, in standard conditions, using an upper arm BP monitor (Omron 705IT, Omron Healthcare, Kyoto, Japan). Ambulatory 24-h BP automated monitoring was performed using a 24-h ambulatory peripheral BP monitor TM2430 (A&D Medical, San Jose, CA, United States). The frequency of measurements was every 15 min between 7:00 and 22:00 and every 30 min between 22:00 and 7:00. Subsequently, mean arterial pressure (MAP) was calculated from the formula MAP = DBP + 1/3 (SBP—DBP) (mm Hg). Mean daily and nocturnal values of SBP and DBP were analyzed. The percentage drop in BP (night blood pressure fall—NBPF) was calculated using the following equation: NBPF = ((MAP day—MAP night)/MAP day) × 100%. Patients with normal NBPF (10–20%) were referred to as “dippers.” Patients with NBPF < 10% were classified as “non-dippers,” patients with NBPF exceeding 20% were classified as “extreme dippers,” while patients with elevation of BP at night were “reverse-dippers” [18,19,20].

#### 2.4.2. Neck Circumference Measurement

The neck circumference was measured in the midway of the neck, between the mid-cervical spine and mid-anterior neck, in standing position, with a flexible non-stretchable plastic tape, and approximated to the nearest 0.1 cm.

#### 2.4.3. Echocardiographic Measurement

All patients underwent complete transthoracic echocardiographic study with Vivid S6 instrument (GE Medical System, Tirat Carmel, Israel) with a 1.5–3.6-MHz matrix cardiac sector probe. A standard M-mode, two-dimensional and Doppler echocardiographic examination was performed according to the guidelines of American Society of Echocardiography [21,22]. Three consecutive cycles were averaged for every parameter. The same experienced cardiologist who was blinded to the presence or absence the additional therapy of eplerenone, performed all echocardiographic examinations. Left ventricular end-diastolic diameter (LVED), thickness of intraventricular septum at end diastole (IVS), left ventricular posterior wall at end diastole (LVPW), and left ventricular mass (LVM) were measured according to The American Society of Echocardiography recommendations [21]. The LVH was defined as the IVS or the LVPW ≥ 12 mm. The LVM was calculated using a simple and anatomically validated formula:LVM = 0.8 × 1.04 ((IVS + LVEDD + LVPW) 3 − LVEDD3) + 0.6

LVM was calculated as corrected for height and LVM index (LVMI) [22]. The relative wall thickness (RWT) was calculated as (2 × LVPW)/LVEDD, for which the normal limit is <0.42 [21,22]. Based on LVMI and relative wall thickness (RWT), the LV geometry was classified as normal (LVMI < 115 g/m^2^ in men and <95 g/m^2^ in women, and RWT < 0.42), concentric remodeling (CR) (normal LVMI < 115 g/m^2^ in men and <95 g/m^2^ in women, and increased RWT > 0.42), concentric hypertrophy (CH) (LVMI > 115 g/m^2^ in men and >95 g/m^2^ in women, and increased RWT > 0.42) or eccentric hypertrophy (EH) (LVMI > 115 g/m^2^ in men and >95 g/m^2^ in women, and normal RWT < 0.42) [23].

#### 2.4.4. Polysomnography (PSG)

The probability of OSA was established at first on the base of Epworth Sleepiness Scale score [24]. The evaluation of patients was performed in the Sleep Laboratory of the Department of Pulmonology, Allergology and Respiratory Oncology at the University of Medical Sciences in Poznan, Poland using a full-night polysomnographic monitoring system (EMBLA S4000, Remlogic, Denver, CA, USA) with Somnologica studio 3.3.2 software (EMBLA, Broomfield, CA, USA). Standard electroencephalography monitoring, including frontal leads (F1, F2), central leads (C3, C4), occipital leads (O1, O2), and reference leads at the mastoids (M1, M2); electromyography; and electrooculography methodology were performed according to The American Academy of Sleep Medicine (AASM) guidelines [25,26]. Airflow was measured using nasal thermistors and a nasal pressure transducer. Abdominal and thoracic movements were assessed by respiratory inductive plethysmography. Oximetry was measured using a disposable finger probe (oximeter flex sensor 8000 J, NONIN, Plymouth, MN, USA) placed on the index finger. Snoring sounds and heart rate were also recorded. Body position was monitored using body position sensor. Apnea was defined as a cessation of airflow lasting for more than 10 s, and hypopnea as a discrete reduction (two-thirds) of airflow and/or abdominal ribcage movements lasting for more than 10 s and associated with a decrease of more than 4% in oxygen saturation. The apnea–hypopnea index (AHI) was defined by the total number of apneas and hypopneas per hour of sleep. The severity of OSA was determined on the basis of AHI as: mild (AHI 5–14), moderate (AHI 15–29), and severe (AHI ≥ 30) [27,28].

### 2.5. Statistical Analysis

Statistical analyses were performed with Statistica, v. 12.5 (StatSoft, Tulsa, OK, USA). Since the tested data did not meet the assumption of Gaussian distribution (evaluated with Shapiro–Wilk method), the non-parametric methods were applied to all analyses. Differences between matched samples (parameters at Visit 1 and Visit 2) were evaluated with Wilcoxon signed-rank test, differences in parameters between eplerenone add-on and standard antihypertensive therapy group were tested with Mann–Whitney *U* test. The differences in frequencies of patients classified to different sub-groups were evaluated with Pearson’s chi-square test. In all analyses, a *p* < 0.05 was considered as statistically significant.

## 3. Results

### 3.1. Studied Group

The general characteristics of examined groups are presented in Table 1. The study involved 51 patients (29 males and 22 females) with median age of 59 years assigned to Group 1—add-on 6-month therapy with eplerenone (50 mg/daily) and to Group 2 (30 males and 21 females) receiving standard antihypertensive therapy. All enrolled patients in both groups successfully completed the study.

### 3.2. Body Composition Parameters

The group completing the 6-month add-on therapy with eplerenone revealed decreased values of BMI (by 2.8%), weight (by 1.1%), neck circumference (by 2.3%), and waist circumference (by 0.9%) compared to the baseline levels. None of these parameters differed significantly from the standard antihypertensive group (Table 1).

### 3.3. Renal and Biochemical Parameters

Renal and biochemical parameters monitored during Visit 1 and Visit 2 for the patients assigned to add-on treatment with eplerenone and standard antihypertensive group are summarized in Table 1. Compared to Visit 1, the group receiving eplerenone exhibited a 6.5-fold, statistically significant increase in plasma renin activity after 6 months, and that increase was higher than in the standard antihypertensive group. Moreover, compared to Visit 1 and to the standard antihypertensive group, patients receiving eplerenone treatment revealed significantly decreased plasma aldosterone concentration (by 45.5 and 46.1%) and lower aldosterone–renin ratio (by 91 and 55.6%, respectively). Glomerular filtration rate was slightly but significantly lower in both studied groups; however, no relevant intergroup differences were noted. The Group 1 receiving eplerenone add-on therapy displayed significantly increased serum potassium concentration after 6 months (by 7.3% in both cases—compared to the baseline data from Visit 1 and to results achieved in Group 2 (Table 1).

### 3.4. Blood Pressure Parameters

The BP parameters at Visit 1 and Visit 2 for both studied groups are presented in Table 2. Compared to Visit 1 and to the standard antihypertensive group, all monitored BP parameters were significantly reduced after 6-month add-on therapy with eplerenone. The greatest decline was observed for night-time BP, (SBP, DBP, and MAP were decreased by 14.4, 6.4, and 9.5 mm Hg, respectively) (Figure 2). Therefore, the NBPF in this group has increased from 4.6% to 8.9%, and this also resulted in a reduction in the number of non-dipper patients by 23 patients in this group (Table 3). In the group without eplerenone, no changes in the circadian BP profile were observed.

Patients in group with eplerenone add-on therapy were divided into two subgroups: with plasma aldosterone concentration (PAC) ≥ 15 ng/mL and <15 ng/mL. The greater drop in blood pressure (SBP, DBP, MAP) was recorded at night in the subgroup with PAC ≥ 15 ng/mL (*p* < 0.05 in all cases; Mann–Whitney *U* test) (Figure 3).

### 3.5. Polysomnographic and Echocardiographic Parameters

Polysomnographic and echocardiographic parameters and their statistical comparisons for both studied groups are summarized in Table 4. There was a significant decrease in AHI in the group receiving eplerenone add-on therapy compared to baseline and to standard antihypertensive group (by 34.5% in both groups). Mean and lowest saturation levels were significantly increased after 6-month add-on treatment with eplerenone (by 3.4 and 4.1%, respectively) but did not differ from standard antihypertensive group (Table 4). A greater decrease in BP, particularly systolic at night in the group with the highest AHI values has been demonstrated. In patients receiving eplerenone, significant positive correlations between change in AHI and change in LVEDD, LVMI, and LVM were found (Figure 4).

Additionally, a positive correlation (Rs = 0.5) between the change in PAC and change in AHI was found in this group (Figure 4). Compared to baseline and to Group 2, the significant decrease in IVS (by 6.8 and 7.3%, respectively), PWd (by 6.6 and 6.9%, respectively), LVMI (by 9.4 and 10.5%, respectively), and RWT (by 3.9 and 5.8%, respectively) were noted in the group receiving eplerenone. However, this group also revealed significantly decreased LVEDD after 6 months (by 2.4%), and no statistical difference with standard antihypertensive group was found (Table 4). As shown in Figure 5, the proportion of patients with concentric remodeling increased from 43% to 72% (*p* < 0.05) while the frequency of concentric hypertrophy significantly decreased from 49% to 18% (*p* < 0.05). In turn, no changes in geometric patterns were observed in the standard antihypertensive group (Figure 5).

Number of correlations between changes in echocardiographic parameters and BP parameters were found in both groups after 6 months of observation. There was, however, no correlation between change in LVEDD, IVSd, PWd, LVMI, LVM, and RWT, and change in NBPF (Table 5).

### 3.6. Side Effects

No adverse events were recorded in both groups.

## 4. Discussion

This study is the first to report the improvement of the circadian BP profile and enhancement of the left ventricle geometry in the group of patients with RAH and OSA after the addition of selective MRA—eplerenone. There are numerous studies on the clinical benefits of the use of spironolactone in this group of patients; however, no study addressed the effect of eplerenone in this specific but common group of patients [4]. Compared to spironolactone, eplerenone is a newer and more selective MRA. As demonstrated, it reveals the lower risk of sex-related adverse effects [29]. Eplerenone lowers BP, inhibits heart muscle fibrosis, and may cause a release of endothelial nitric oxide and attenuate systemic oxidative stress. Antihypertensive effect achieved by this compound is mediated mainly by the reduction of fluid retention. It is likely that in patients with OSA, a reduction of fluid accumulation, especially at the level of the neck, may contribute to the lowering of the resistance in the upper respiratory tract, and eventually be helpful in decreasing the severity of OSA.

As demonstrated in the present study, addition of eplerenone to the standard antihypertensive therapy resulted in the BP drop in the office and ABPM measurements. The greatest decrease was observed in the night hours and it included systolic, diastolic and mean blood pressure. This relation is supported by results of previous studies on this matter in which the use of older non-selective MRA—spironolactone—reduced the severity of OSA by reducing AHI, as well as caused a significant decrease in day-time and night-time BP. Gaddam et al. obtained a significant BP and AHI reduction after 8 weeks of treatment with 50 mg/day of spironolactone, while Yang et al. observed a statistically significant decrease in AHI and plasma aldosterone concentration in patients with OSA and RAH after 12 weeks treatment [12,30]. Similar findings were also reported by Kasai et al. [31]. In the present study, a reduction in the AHI index from 44/h to 28/h was achieved in the eplerenone treated group, as already described in the previous paper [32]. As demonstrated in the present study, the greatest blood pressure drop in SBP, DBP, and MAP at night-time were obtained after eplerenone administration in the sub-group with higher PAC (>15) and AHI (>30/h). Moreover, this group also revealed a positive, favorable correlation between the decrease in PAC and the decrease in AHI between Visits 1 and 2. Ke et al. presented similar correlations in their latest analysis of multiple linear regressions on a group of 534 patients with RAH. They showed that patients with AHI ≥ 30 had a higher PAC and aldosterone level in a 24-h urine collection [33]. In the present study, treatment with eplerenone resulted in a beneficial increase in the excretion of sodium and fluids, which could be the cause for the reduction of upper airway obstruction and of AHI. It appears that in patients with OSA, the aldosterone antagonists, apart from acting directly as a diuretic, also act indirectly by reducing the swelling of the back of the throat and the tongue. This diminishes the size of the collapse of the upper airway tract during sleep and supports the treatment of OSA.

In addition, intensified diuretic therapy could reduce the overnight change in leg fluid volume with return of fluids to vascular system which eventually results in lower risk of peripheral edema also at the level of the neck, as well as lower BP values at night regardless from change in body position. On the other hand, eplerenone also prevents the activation of the renin–angiotensin–aldosterone system, which is activated by intermittent hypoxemia. In our study, only the eplerenone treated group revealed a greater BP drop at night, which improved the 24-h patient BP profile. Eplerenone reduced the number of non-dipper patients from 80% to 35% and the NBPF increased from 5.7% to 8.9%, which approached favorably to the desired value of 10%. This observation may be of clinical significance because patients in whom the NBPF does not exceed 10% (non-dippers) are characterized by a higher risk of developing subclinical organ damage. Verdecchia et al. were the first who showed that non-dipper hypertensive patients had 3-fold higher frequency of cardiovascular events, compared to patients with dipper BP profile [34]. As demonstrated in the Syst-Eur study, higher incidence of stroke and myocardial infraction was revealed in non-dipper patients [35]. In patients with circadian BP profile disorders such complications like LVH, microalbuminuria, thickening of the intima media in the carotid artery and neurological incidents are significantly more common [36,37,38]. Finding of increased incidence of concentric hypertrophy (CH) may partially explain the higher incidence of cardiovascular events in these patients. In our population of patients with RAH and OSA, as many as 38 (74.5%) of them had been diagnosed with LVH (IVS or LVPW ≥ 12 mm). After 6 months of add-on therapy with eplerenone, a significant decrease in the values of echocardiographic parameters describing dimension, thickness, and mass of the left ventricle were revealed. This resulted in the improvement of left ventricular geometry and in the reduction of the number of patients with concentric hypertrophy from 49% to 18% in favor of the geometry of concentric remodeling (CR). In the eplerenone group, we showed a positive, favorable correlation between the reduction of the LVM index and AHI index when comparing results from Visits 1 and 2. This relationship was confirmed also in a large cross-sectional study—Sleep Heart Health Study. The LVMI index significantly correlated with both AHI and the hypoxaemia index [9]. Similarly, Koga et al. showed in their analyses that the AHI was an independent factor associated with the presence of concentric hypertrophy [39]. In another study, the authors demonstrated significant associations between severity of OSA and left ventricle structure using cardiac magnetic resonance imaging. Independent of confounders, higher levels of AHI were significantly associated with increased left ventricle mass [40]. Some studies reported that eccentric hypertrophy is most closely associated with OSA, whereas others, including our results, describe that in patients with OSA, the concentric hypertrophy is most common [41,42]. Cioffi et al. found a high prevalence of CH in patients with moderate/severe OSA and RAH, as well as a positive association between left ventricular relative wall thickness and severity of OSA [8]. Koga et al. found that patients with RAH and OSA showed a high prevalence of concentric LVH patterns, which was diminished after 3 months of CPAP treatment [39]. The researchers argued that OSA (intermittent hypoxemia and recurrent arousals during sleep) contributes to the development of LVH, probably through changes in left ventricular afterload [43,44]. In the current study, after adding eplerenone to standard antihypertensive therapy, we obtained a positive correlation between the reduction of left ventricular end-diastolic dimension (LVEDD), LVM, and LVMI parameters and the reduction of SBP during daily activity in ABPM and the negative relationship between reduction of LVEDD dimension and increase in the RWT index. We did not find a correlation between the decrease of BP at night-time and reduction of LVMI, despite the fact that the drop of SBP, DBP, and MAP was the greatest at night-time.

In addition to the correlation described above, other relations between NBPF and decrease in LVH were statistically insignificant and values of those parameters did not differ between the groups with and without eplerenone. Moreover, this correlation remained statistically insignificant even after dividing patients into non-dipper, dipper, and extreme dipper subgroups. It could indicate that LVH occurs in this group of patients regardless of the dippers’ or non-dippers’ profile, i.e., regardless of elevated BP at night or day. This lack of correlation between decrease in the night-time BP and a decrease in LVH may also indicate that this relationship is not limited to only one marker but can be influenced by many different related factors. In the literature, this relationship between high night-time BP, i.e., the non-dippers’ profile, and LVH, which is an important marker of organ damage, is unclear. There are studies that prove that the occurrence of CH is related to increased sympathetic nervous system activity and elevated BP at night, and this is associated with the non-dippers’ profile that often accompanies OSA patients [23,45]. Already in 1995, Fagart published a contradictory meta-analysis, which demonstrated that both day and night arterial hypertension is an LVH indicator with a weak or no correlation between LVM and a day–night difference in BP. In 19 studies involving 2223 patients with normal BP or hypertension, night-time hypertension was not a better predictor of LVM than day-time hypertension [46]. In a review paper from 2010, Cuspidi et al. analyzed 26 studies published in the last decade: in 17 studies (a total of 2497 patients), a positive correlation was found between the non-dippers and LVH, while in the remaining 9 this relationship was not obtained [15].

## 5. Limitations

The study provides evidence that eplerenone is a valuable addition to antihypertensive treatment in patients with RAH and OSA; however, some limitations need to be stressed. Firstly, the observations require further confirmation on larger sample size. The clinical trial was designed for 6 months, while changes in some parameters such as drop in BP at night and decrease in LVH may require longer period to occur. Secondly, the present study enrolled a specific group with RAH and OSA; thus the results cannot be extrapolated to entire population of patients revealing RAH. Thirdly, it is known that in general, LV hypertrophy can regress after a decrease in BP, brought about by appropriate hypotensive medication, and also after a decrease in BMI. In our study, there was a small but significant weight loss of 1.1 kg in studied population, which could influence the achieved results. Due to the fact that this decline could have been caused by numerous factors and the fact that such result was not expected initially, we did not use more relevant tools (questionnaires, assessment of edema, etc.) to assess subject of weight and diet in detail in our study—we decided not to speculate about it in discussion part. The antihypertensive treatment was equally successful in both groups in maintaining the BP control at baseline, although no limitations were made on the type of drugs used or their dosing except of aldosterone receptor blocker.

## 6. Conclusions

The present study demonstrated benefits of eplerenone as an add-on 6-month therapy in the reduction in blood pressure parameters as well as in improving the circadian blood pressure profile in patients with OSA and RAH. Moreover, eplerenone had a positive implication on polysomnographic and echocardiographic parameters and a correlation between them. The results of this study have shown that the addition of the aldosterone antagonist to a standard antihypertensive therapy reduces LVH, and thus may lead to better cardiovascular outcomes in patients with OSA and RAH. This group of antihypertensive drugs could be an effective therapeutic option, without side effects for this group of patients. The study emphasizes the importance of screening for OSA in patients with resistant hypertension and highlights the potential benefit of eplerenone in treatment in this population.

## Figures and Tables

**Figure 1 jcm-08-01671-f001:**
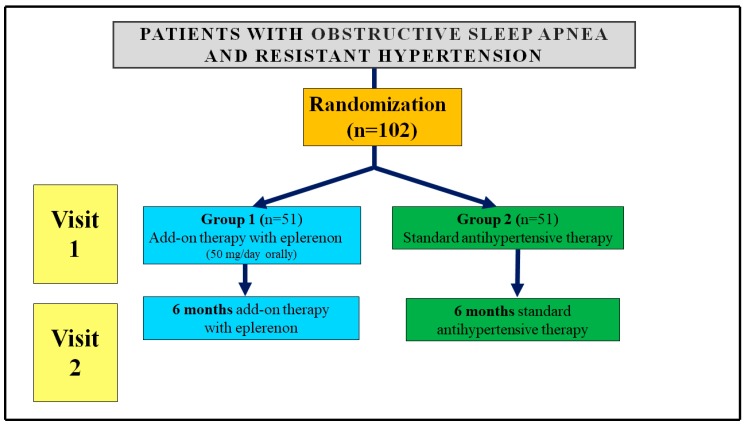
Scheme of the study.

**Figure 2 jcm-08-01671-f002:**
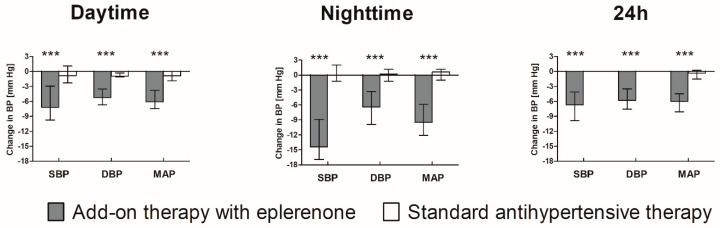
Changes in BP parameters during 6-month add-on treatment with eplerenone compared to standard antihypertensive therapy. Asterisks (***) denote statistically significant differences between groups (Mann–Whitney *U* test). SBP—systolic blood pressure; DBD—diastolic blood pressure; MAP—mean arterial pressure.

**Figure 3 jcm-08-01671-f003:**
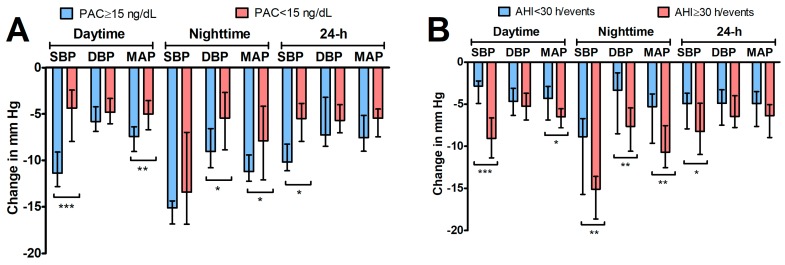
Changes in BP parameters (median and interquartile range) in patients receiving eplerenone for 6 weeks (*n* = 51) in relation to their baseline aldosteronism (**A**) and apnea–hypopnea index (AHI) (**B**)status. Asterisks indicate significant differences (* *p* < 0.05; ** *p* < 0.01; *** *p* < 0.001; Mann–Whitney *U* test).

**Figure 4 jcm-08-01671-f004:**
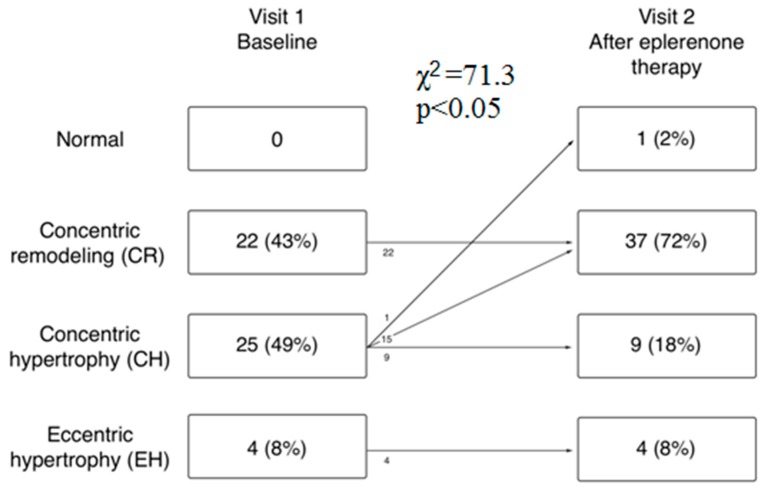
Left ventricular geometric patterns in patients with obstructive sleep apnea syndrome and resistant hypertension before (Visit 1) and after 6 months of eplerenone add-on therapy (Visit 2). No changes in geometric patterns were observed in the placebo group.

**Figure 5 jcm-08-01671-f005:**
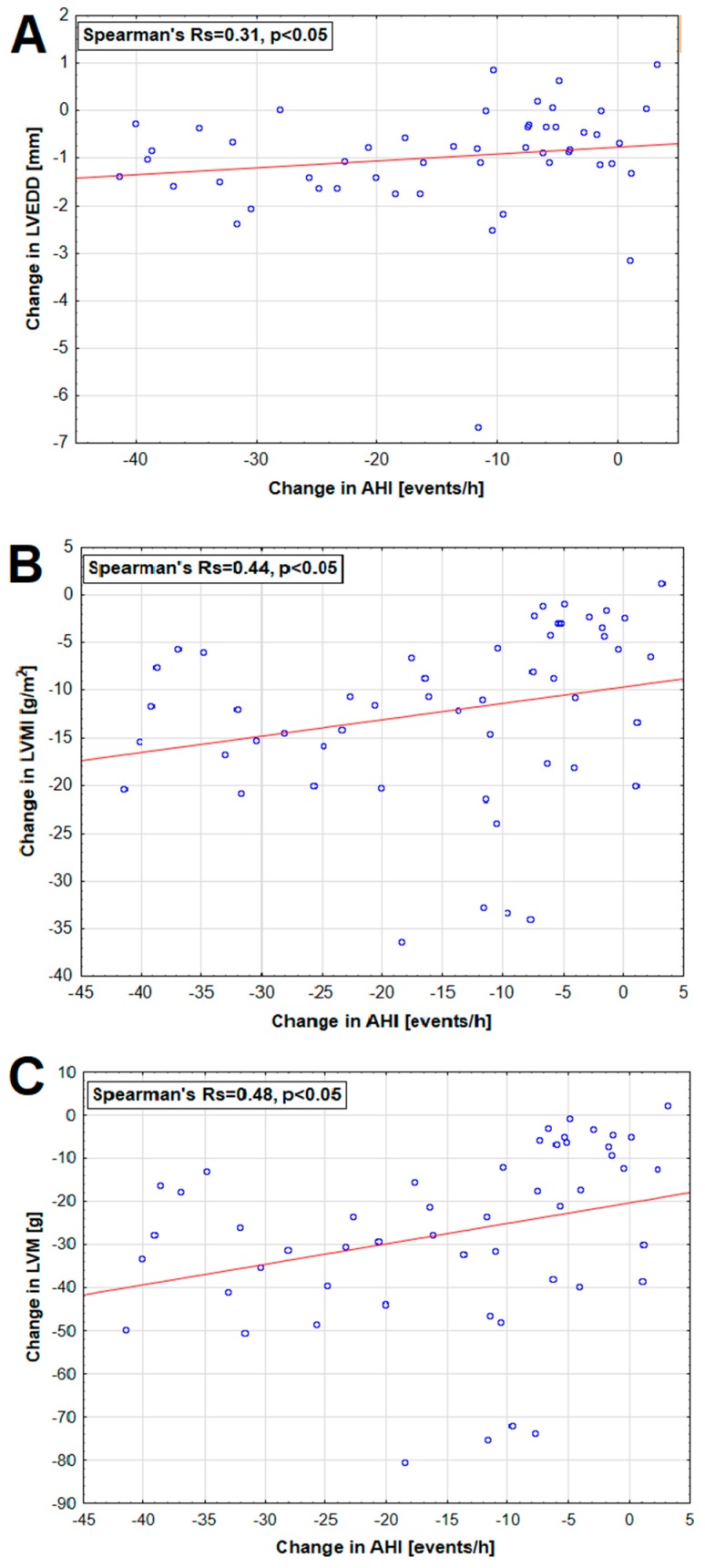
Relationship between change (Visit 1 and Visit 2) in AHI and change in (**A**) LVEDD, (**B**) LVMI, and (**C**) left ventricular mass (LVM) in patients receiving eplerenone.

**Table 1 jcm-08-01671-t001:** Demographical characteristics, body composition, and renal parameters of patients assigned to add-on treatment with eplerenone (50 mg/day) and standard hypertensive treatment before (V1) and after 6 months (V2). All values are reported as median (interquartile range). The statistical comparison of parameters before (V1) and after treatment (V2) was conducted with Wilcoxon test (WLCXN) while the Eplerenone versus standard hypertensive treatment before comparison was conducted with Mann–Whitney *U* test (MW-*U*) except Pearson’s chi-squared test (χ^2^) for gender.

Parameter		Add-on Therapy—with Eplerenone	Standard Antihypertensive Therapy	
Gender (*n*/%)	**V1**			*p* > 0.05
M		29	30	
F		22	21	
Age (years)	**V1**	59.0 (52.0–62.0)	59.0 (52.0–62.0)	*p* > 0.05
Height (cm)	**V1**	1.7 (1.64–1.75)	1.7 (1.65–1.76)	*p* > 0.05
Weight (kg)	**V1**	102.3 (88.7–112.4)	102.3 (90.3–111.2)	*p* > 0.05
	**V2**	101.2 (87.7–111.9)	101.3 (83.4–116.9)	*p* > 0.05
	WLCXN	*p* < 0.001	*p* > 0.05	
Body mass index (kg/m^2^)	**V1**	36.1 (33.0–37.6)	35.4 (32.1–37.4)	*p* > 0.05
	**V2**	35.6 (32.3–37.2)	35.3 (30.3–39.1)	*p* > 0.05
	WLCXN	*p* < 0.001	*p* > 0.05	
Neck (cm)	**V1**	43.0 (40.0–47.0)	44.0 (39.0–46.0)	*p* > 0.05
	**V2**	42.0 (39.0–44.0)	44.0 (34.0–47.0)	*p* > 0.05
	WLCXN	*p* < 0.001	*p* > 0.05	
Waist (cm)	**V1**	105.0 (98.0–118.0)	104.0 (98.0–116.0)	*p* > 0.05
	**V2**	104.0 (99.0–116.0)	103.0 (87.0–121.0)	*p* > 0.05
	WLCXN	*p* < 0.001	*p* > 0.05	
Serum potassium (mmol/L)	**V1**	4.1 (3.9–4.3)	4.1 (3.8–4.6)	*p* > 0.05
	**V2**	4.4 (4.2–4.6)	4.1 (3.5–4.6)	*p* > 0.001
	WLCXN	*p* < 0.001	*p* > 0.05	
Serum creatinine (µmol/L)	**V1**	82.4 (73.4–90.9)	80.1 (74.6–89.3)	*p* > 0.05
	**V2**	85.7 (77.5–92.4)	80.9 (74.3–91.3)	*p* > 0.05
	WLCXN	*p* < 0.01	*p* < 0.05	
Glomerular filtration rate (mL/min)	**V1**	85.0 (62.0–99.0)	81.0 (63.0–96.0)	*p* > 0.05
	**V2**	80.0 (62.0–96.0)	80.0 (63.0–96.0)	*p* > 0.05
	WLCXN	*p* < 0.01	*p* < 0.05	
Plasma renin activity (ng/mL/h)	**V1**	0.2 (0.2–1.3)	1.0 (0.2–1.2)	*p* < 0.01
	**V2**	1.3 (1.1–2.1)	1.2 (0.2–1.8)	*p* < 0.001
	WLCXN	*p* < 0.001	*p* < 0.05	
Plasma aldosterone (ng/dL)	**V1**	10.1 (6.5–12.9)	10.3 (7.4–11.3)	*p* > 0.05
	**V2**	5.5 (3.5–7.3)	10.2 (5.7–13.5)	*p* < 0.001
	WLCXN	*p* < 0.001	*p* > 0.05	
Aldosterone–renin ratio	**V1**	4.8 (0.4–8.6)	1.0 (0.7–5.3)	*p* > 0.05
	**V2**	0.4 (0.2–0.6)	0.9 (0.5–5.2)	*p* < 0.001
		*p* < 0.001	*p* < 0.05	

**Table 2 jcm-08-01671-t002:** The BP parameters (mm Hg) observed for patients assigned to add-on treatment with eplerenone (50 mg/day) and standard hypertensive treatment before (V1) and after 6 months (V2). All values are reported as median (interquartile range). The statistical comparison of parameters before (V1) and after treatment (V2) were conducted with Wilcoxon test (WLCXN) while the eplerenone versus standard hypertensive treatment comparison was conducted with Mann–Whitney *U* test (MW-*U*).

		Add-On Therapy—with Eplerenone	Standard Antihypertensive Therapy	
cSBP	**V1**	150.0 (144.0–155.0)	148.0 (145.0–155.0)	*p* > 0.05
	**V2**	142.0 (138.0–145.0)	149 (144.0–153.0)	*p* < 0.001
	WLCXN	*p* < 0.001	*p* > 0.05	
cDBP	**V1**	92.0 (89.0–99.0)	91.0 (90.0–98.0)	*p* > 0.05
	**V2**	90.0 (84.0–93.0)	91.0 (88.0–98.0)	*p* < 0.001
	WLCXN	*p* < 0.001	*p* > 0.05	
SBPd	**V1**	147.7 (141.2–155.2)	147.6 (142.3–154.3)	*p* > 0.05
	**V2**	140.3 (137.7–147.5)	146.6 (141.3–152.3)	*p* < 0.001
	WLCXN	*p* < 0.001	*p* > 0.05	
DBPd	**V1**	96.5 (90.2–101.0)	96.4 (92.3–99.8)	*p* > 0.05
	**V2**	90.3 (86.2–94.7)	95.5 (91.8–99.5)	*p* < 0.001
	WLCXN	*p* < 0.001	*p* > 0.05	
MAPd	**V1**	112.8 (109.2–117.6)	112.9 (110.7–116.6)	*p* > 0.05
	**V2**	107.5 (103.7–110.3)	112.8 (109.7–116.1)	*p* < 0.001
	WLCXN	*p* < 0.001	*p* > 0.05	
SBPn	**V1**	141.0 (132.1–146.8)	136.7 (132.0–140.3)	*p* > 0.05
	**V2**	125.4 (121.0–131.6)	136.5 (133.3–140.4)	*p* < 0.001
	WLCXN	*p* < 0.001	*p* > 0.05	
DBPn	**V1**	91.2 (86.7–97.8)	90.1 (85.8–93.2)	*p* > 0.05
	**V2**	84.4 (80.2–89.8)	90.3 (86.6–92.3)	*p* < 0.001
	WLCXN	*p* < 0.001	*p* > 0.05	
MAPn	**V1**	108.3 (103.1–114.6)	104.5 (102.0–107.6)	*p* > 0.05
	**V2**	98.2 (94.6–103.1)	104.9 (102.0–107.4)	*p* < 0.001
	WLCXN	*p* < 0.001	*p* > 0.05	
SBP-24	**V1**	142.9 (139.4–151.2)	143.3 (140.4–150.3)	*p* > 0.05
	**V2**	138.5 (133.3–142.2)	142.8 (139.4–149.0)	*p* < 0.001
	WLCXN	*p* < 0.001	*p* > 0.05	
DBP-24	**V1**	94.3 (88.8–99.9)	93.5 (90.8–97.8)	*p* > 0.05
	**V2**	88.7 (82.6–92.5)	94.0 (91.1–97.8)	*p* < 0.001
	WLCXN	*p* < 0.001	*p* > 0.05	
MAP-24	**V1**	109.9 (106.9–116.6)	111.0 (109.1–114.3)	*p* > 0.05
	**V2**	104.8 (101.3–108.0)	111.0 (107.6–113.2)	*p* < 0.001
	WLCXN	*p* < 0.001	*p* > 0.05	
NBPF (%)	**V1**	4.6 (2.9–6.2)	7.7 (6.7–10.0)	*p* < 0.01
	**V2**	8.9 (6.7–10.0)	7.4 (4.1–10.1)	*p* > 0.05
	WLCXN	*p* < 0.001	*p* > 0.05	

cSBP, clinic systolic blood pressure; cDBP, clinic diastolic blood pressure; SBPd, ambulatory day-time systolic blood pressure; DBPd, ambulatory day-time diastolic blood pressure; MAPd, ambulatory day-time mean blood pressure; SBPn, ambulatory night time systolic blood pressure; DBPn, ambulatory night time diastolic blood pressure; MAPn, ambulatory night time mean blood pressure; SBP24, 24 h systolic blood pressure; DBP24, 24-h diastolic blood pressure; MAP24, 24-h mean blood pressure; NBPF, night blood pressure fall.

**Table 3 jcm-08-01671-t003:** Frequency (*n*/%) of non-dippers, dippers, and extreme dippers in the group with add-on treatment with eplerenone (50 mg/day) and standard antihypertensive treatment before (V1) and after 6 months (V2).

	V1		V2	
	Add-On Therapy—with Eplerenone	Standard Antihypertensive Therapy	Add-On Therapy—with Eplerenone	Standard Antihypertensive Therapy
**Non-Dippers**	41/80.4	38/74.5	18/35.3	39/76.5
**Dippers**	10/19.6	10/19.6	31/60.8	9/17.6
**Extreme Dippers**	0/0.0	3/5.9	2 (3.9)	3/5.9
	χ^2^ = 3.2, *p* > 0.05		χ^2^ = 20.0, *p* < 0.001	

**Table 4 jcm-08-01671-t004:** The polysomnographic and echocardiographic parameters observed for patients assigned to add-on treatment with eplerenone (50 mg/day) and standard hypertensive treatment before (V1) and after 6 months (V2). All values are reported as median (interquartile range). The statistical comparison of parameters before (V1) and after treatment (V2) were conducted with Wilcoxon test (WLCXN) while the eplerenone versus standard hypertensive treatment comparison was conducted with Mann–Whitney *U* test (MW-*U*).

Parameter		Add-On Therapy—with Eplerenone	Standard Antihypertensive Therapy	MW-*U* test
AHI (/h)	**V1**	44.0 (25.6–61.4)	45.9 (28.4–61.3)	*p* > 0.05
	**V2**	28.8 (22.3–38.5)	44.0 (29.0–60.0)	*p* < 0.001
	WLCXN	*p* < 0.001	*p* > 0.05	
AI (/h)	**V1**	33.9 (14.4–46.7)	33.9 (14.4–46.7)	*p* > 0.05
	**V2**	33.5 (17.6–45.8)	37.0 (20.9–54.6)	*p* > 0.05
	WLCXN	p>0.05	*p* > 0.05	
ODI (/h)	**V1**	23.0 (14.2–54.3)	22.4 (13.3–49.6)	*p* > 0.05
	**V2**	28.5 (15.2–55.7)	34.0 (15.0–57.0)	*p* > 0.05
	WLCXN	*p* < 0.001	*p* < 0.05	
Mean saturation (%)	**V1**	89.2 (86.0–92.4)	90.3 (88.8–91.3)	*p* > 0.05
	**V2**	92.2 (90.2–93.5)	89.8 (87.6–91.3)	*p* < 0.001
	WLCXN	*p* < 0.001	*p* > 0.05	
Lowest saturation (%)	**V1**	72.4 (59.0–82.3)	69.3 (61.3–80.3)	*p* > 0.05
	**V2**	75.4 (66.6–79.7)	66.8 (62.4–80.3)	*p* > 0.05
	WLCXN	*p* < 0.001	*p* > 0.05	
LVEDD (cm)	**V1**	49.9 (45.7–55.7)	49.9 (45.6–53.3)	*p* > 0.05
	**V2**	48.7 (45.2–54.4)	49.9 (45.3–53.2)	*p* > 0.05
	WLCXN	*p* < 0.001	*p* > 0.05	
IVS (cm)	**V1**	13.2 (11.4–14.3)	13.1 (11.3–13.9)	*p* > 0.05
	**V2**	12.3 (11.1–13.3)	13.2 (11.4–13.9)	*p* < 0.05
	WLCXN	*p* < 0.001	*p* > 0.05	
PWd (cm)	**V1**	12.9 (11.5–13.5)	13.0 (11.1–13.5)	*p* > 0.05
	**V2**	12.1 (11.1–13.0)	13.0 (1.5–13.7)	*p* < 0.05
	WLCXN	*p* < 0.001	*p* < 0.05	
LVMI (g/m^2^)	**V1**	116.9 (93.8–153.9)	118.7 (87.6–150.9)	*p* > 0.05
	**V2**	105.9 (87.2–141.0)	118.4 (90.8–144.4)	*p* > 0.05
	WLCXN	*p* < 0.001	*p* > 0.05	
RWT	**V1**	0.51 (0.47–0.53)	0.50 (0.47–0.54)	*p* > 0.05
	**V2**	0.49 (0.46–0.52)	0.52 (0.48–0.55)	*p* < 0.05
	WLCXN	*p* < 0.001	*p* < 0.01	

AHI, apnea–hypopnea index; AI, arousal index; ODI, oxygen desaturation index; LVEDD, —left ventricular end-diastolic dimension; IVS, interventricular septum thickness at end-diastole; PWd, posterior wall diameter; LVMI, left ventricular mass indexed to body surface area; RWT, relative wall thickness.

**Table 5 jcm-08-01671-t005:** Relationship (Spearman’s correlation coefficients) between changes in BP (between Visit 1 and 2) and changes in echocardiographic parameters after 6 months in eplerenone-treated and in placebo group. Statistically significant correlations are marked in bold.

	Group	LVEDD	IVSd	PWd	LVMI	LVM	RWT
**Day-Time** **SBP**	EPL	**0.38**	0.14	0.05	**0.29**	**0.32**	−0.17
	SAT	−0.04	−0.01	−0.06	−0.02	−0.01	0.00
**Day-Time DBP**	EPL	0.02	0.08	−0.12	0.02	0.05	−0.12
	SAT	0.26	0.02	0.13	0.18	0.21	−0.06
**Day-Time MAP**	EPL	0.15	0.11	−0.11	0.10	0.13	−0.20
	SAT	0.21	0.02	0.02	0.11	0.14	−0.09
**Night-time SBP**	EPL	0.17	0.12	0.12	0.23	0.24	0.02
	SAT	**−0.32**	0.07	0.03	**−0.29**	**−0.30**	0.17
**Night-Time DBP**	EPL	0.15	0.22	0.09	0.21	0.24	0.02
	SAT	0.18	−0.04	**0.39**	**0.31**	**0.29**	**0.28**
**Night-Time MAP**	EPL	0.19	0.18	0.12	0.23	0.26	0.01
	SAT	−0.05	0.01	**0.36**	0.12	0.11	**0.37**
**24-h SBP**	EPL	**0.34**	0.03	−0.10	0.18	0.20	**−0.28**
	SAT	−0.02	−0.15	−0.07	−0.02	−0.03	−0.10
**24-h DBP**	EPL	−0.08	−0.07	−0.13	−0.10	−0.07	−0.09
	SAT	0.12	−0.10	0.03	−0.05	−0.03	0.02
**24-h MAP**	EPL	0.04	−0.04	−0.15	−0.02	−0.01	−0.17
	SAT	0.06	−0.18	−0.16	−0.13	−0.11	−0.17
**NBPF**	EPL	−0.12	−0.15	−0.17	−0.19	−0.21	−0.12
	SAT	0.10	−0.16	−0.03	−0.03	−0.02	−0.11

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
