# Peer review of "The Effects of Eplerenone on the Circadian Blood Pressure Pattern and Left Ventricular Hypertrophy in Patients with Obstructive Sleep Apnea and Resistant Hypertension—A Randomized, Controlled Trial"

_jcm, 2019, doi:10.3390/jcm8101671_

Round 1

Reviewer 1 Report

I review with great pleasure the following study by Krasińska et al where they evaluate the The effects of eplerenone on the circadian blood pressure pattern and left ventricular hypertrophy in patients with obstructive sleep apnea and resistant hypertension with a randomized, controlled trial, several, methodological aspects to consider.

Abstract: will benefit from English proofing to improve readability.

Introduction: the last sentence regarding spironolactone would be better suited before the study aim.

Experimental section: please clarify is allocation was concealed and how. Please identify qualifying visit. Overall methods would benefit from reorganizing (i.e, inclusion,exclusion, setting baseline visit, assessments, randomization) adding additional elements re the protocol. Aditional questions:

-who performed the screening and diagnosis of RAH and OSA.

Was the Epworth scale a criteria  for inclusion? Was these patients refereed for PSG or research subjects, unclear how currently described.

Where patients or study members blinded?

How was compliance measured ?

Was placebo given?

Any adverse events noted?

Did any of the control group receive the study drug?

Statistical analysis:which data was not normally distributed?  Additional details are needed. Did the study team perform power calculations? Adjust for multiple comparisons, more details are needed.

Results: body composition parameters - do the authors consider these to be clinically significant ?

Table 1 please report actual p values, some values suggest randomization was not successful, how did the authors account for this in the analysis. Please clarify.

Author Response

We are very grateful for your review and  your valuable remarks.

Reviewer 2 Report

In the present paper the authors report the results of a randomized trial in patients with untreated obstructive sleep apnea (OSA) and resistant hypertension evaluating the effect of the mineralocorticoid receptor blocker eplerenone on parameters of 24 hour ambulatory blood pressure, echocardiographic measures, and measures of OSA severity. They included 102 patients: 51 randomized to eplerenone 50 mg/d, and 51 randomized to placebo. After 6 months, there was reduction in all blood pressure measurements, a reduction in LV mass, and a reduction in OSA severity in the eplerenone group but not in the placebo group. Importantly, there was also a reduction in body weight and body mass index in the verum group. Also importantly, patients were not treated with CPAP.

General comment

The results of this trial are very impressive, and one is tempted to say “too good”. There are two features which are hard to understand. First, why did the authors choose to select untreated OSA patients and to leave them untreated throughout the trial? A clinically more relevant scenario would have been to select OSA patients on CPAP therapy and to study the effect of eplerenone in these patients. Second, the reduction in body weight is surprising as this is not an expected effect of eplerenone therapy. This reduction in weight may have driven at least in part the other effects.

Specific comments

Inclusion criteria: What was the AHI cut-off required for inclusion into the study? How many patients were screened with PSG but not included?

Study procedures: how were the patients instructed regarding diet and physical activity? Any differences between groups?

Echo: were parameters of diastolic LV function assessed?

After the echo methods section the title “2.4.1. Blood pressure” is erroneously written again.

Results and Table 1: in the verum group a reduction in body weight from 102.3 to 101.2 kg is significant with a p value <0.001 while a reduction in weight from 102.3 to 101.3 kg is not (p>0.05). How can this be?

Table 4: the reduction in AHI by eplerenone is extremely suprising

References

The authors included some relatively old papers in the list of references. A relatively recent review on the cardiovascular features of OSA for the introduction section would be Maeder MT et al. Vasc Health Risk Manag 2016

Author Response

(The authors gave the same response as above.)

Reviewer 3 Report

This manuscript described a study of the effect of eplerenon on the blood pressure, polysomnography and echocardiography in patients with obstructive sleep apnea (OSA) and resistant hypertension (RAH). The researchers found the 6 months treatment with eplerenon in comparison with the regular antihypertensive therapy significantly improved the night blood pressure, reduced the number of non-dipper patients and decreased the left ventricular hypertrophy. The study is interesting, and experimental approach is appropriate. However, this group has published a very similar study before(1). In this previous study, they assessed the effect of the same compound eplerenone on the OSA and arterial stiffness in patients with RAH by performing the same measurements as described in this study during a 3 months study. Additionally, they draw almost the same conclusion, that eplerenone significant approved the OSA and RAH conditions. I did not see any new findings above the previous study. So based on this, I think this study can not accept in current form.

Reference:

(1) Krasinska, B., A. Miazga, S. Cofta, et al., Effect of eplerenone on the severity of obstructive sleep apnea and arterial stiffness in patients with resistant arterial hypertension. Pol Arch Med Wewn, 2016. 126(5): 330-9.   

Author Response

(The authors gave the same response as above.)

Round 2

Reviewer 1 Report

I thank the authors for integrating my comments it has improve the clarity of the study methods.

Reviewer 3 Report

All my questions have been answered appropriately.